# MathOdyssey: Benchmarking Mathematical Problem-Solving Skills in Large Language Models Using Odyssey Math Data

## Abstract

Large language models (LLMs) have significantly advanced natural language understanding and demonstrated strong problem-solving abilities. Despite these successes, most LLMs still struggle with solving mathematical problems due to the intricate reasoning required. This paper investigates the mathematical problem-solving capabilities of LLMs using the newly developed "MathOdyssey" dataset. The dataset includes diverse mathematical problems at high school and university levels, created by experts from notable institutions to rigorously test LLMs in advanced problem-solving scenarios and cover a wider range of subject areas. By providing the MathOdyssey dataset as a resource to the AI community, we aim to contribute to the understanding and improvement of AI capabilities in complex mathematical problem-solving. We conduct benchmarking on open-source models, such as Llama-3, and closed-source models from the GPT series and Gemini models. Our results indicate that while LLMs perform well on routine and moderately difficult tasks, they face significant challenges with Olympiad-level problems and complex university-level questions. Our analysis shows a narrowing performance gap between open-source and closed-source models, yet substantial challenges remain, particularly with the most demanding problems. This study highlights the ongoing need for research to enhance the mathematical reasoning of LLMs. The dataset, results, and evaluation code are publicly available [1].

## 1 Introduction

Large language models (LLMs) have demonstrated exceptional proficiency in mastering human language and handling mathematical problems, including typical routine math problems (OpenAI, 2023; Touvron et al., 2023; Reid et al., 2024). In recent years, several benchmarks related to mathematics have been proposed, such as the GSM8K dataset (Cobbe et al., 2021), the MATH dataset (Hendrycks et al., 2021b) and so on. Recent LLMs and prompting approaches have addressed these problems with notable success (OpenAI, 2023; Touvron et al., 2023). For instance, GPT-4, using advanced prompting techniques (OpenAI, 2023), has achieved more than a 90% success rate on GSM8K and 80% on MATH. These achievements indicate that LLMs possess remarkable capabilities in mathematical reasoning.

The quest to improve LLMs' mathematical problem-solving abilities is not just a demonstration of technological advancement but a crucial step toward developing more general and capable artificial intelligence systems. On the one hand, this endeavor requires datasets that accurately measure and challenge the AI's mathematical reasoning beyond basic problems. Although their performance is high on datasets like GSM8K (Cobbe et al., 2021), it remains uncertain how well they handle more complex mathematical challenges, such as those found in university-level courses and competitive high school mathematics. Performance may diminish significantly in these areas. This gap highlights the ongoing need for enhanced mathematical reasoning capabilities in AI, a critical area for assessing cognitive abilities akin to human intelligence. Moreover, a significant obstacle is that many existing datasets might have been included in the training phases of these models, potentially skewing performance metrics. Prominent examples include STEM-Q (Drori et al., 2023), GSM8K

---

[1]https://anonymous.4open.science/r/mathodyssey-C587/

---

**Olympiad-level**

**Problem:** Let $S = \{1, 2, \cdots 2024\}$, if the set of any $n$ pairwise prime numbers in $S$ has at least one prime number, the minimum value of $n$ is ___________.

**Answer:** 16.

**Reasoning:** Taking the 15 numbers $1, 2^2, 3^2, ..., 43^2$. They violate the condition. Furthermore, since S does not contain any non-prime numbers with a minimum prime factor of at least 47 (because $47^2 > 2024$). Set 1 aside, there are only 14 types of non-prime numbers in S, classified by its minimum prime factor. Applying the Pigeonhole Principle, we conclude that n = 16.

---

**High School**

**Problem:** What are the solutions of the quadratic equation $15x^2 = 2x + 8$.

$$\text{A) } \{-\frac{4}{3}, -\frac{3}{2}\} \quad \text{B) } \{-\frac{4}{5}, \frac{2}{3}\} \quad \text{C) } \{-\frac{3}{2}, \frac{4}{5}\} \quad \text{D) } \{-\frac{2}{3}, \frac{4}{5}\}$$

**Answer:** $D$

**Reasoning:** First move all terms to one side: $15x^2 - 2x - 8 = 0$. Then factor into $(5x - 4)(3x + 2) = 0$. Setting $5x - 4$ to zero results in a solution of $x = \frac{4}{5}$ and setting $3x + 2$ to zero results in a solution of $x = -\frac{2}{3}$.

---

**University-level**

**Problem:** Find the limit
$$\lim_{x \to 1} \frac{f(2x^2 + x - 3) - f(0)}{x - 1}$$
given $f'(1) = 2$ and $f'(0) = -1$.

**Answer:** $-5$.

**Reasoning:** Let $g(x) = 2x^2 + x - 3$. Since $g(1) = 0$, the desired limit equals $\lim_{x \to 1} \frac{f(g(x)) - f(g(1))}{x - 1}$. By the definition of the derivative and the chain rule and noting that $g'(1) = 5$, we have

$$\lim_{x \to 1} \frac{f(g(x)) - f(g(1))}{x - 1} = f'(g(1))g'(1) = f'(0)g'(1) = (-1)(5) = -5.$$

---

Table 1: MathOdyssey dataset examples. We demonstrate three distinct levels to challenge various aspects of mathematical knowledge: Olympiad-level, High School, and University-level mathematics. Each example consists of three parts: the problem, the answer, and the reasoning. Note that both GPT-4 Turbo and Llama-3-70B are unable to solve the first Olympiad-level example. See Appendix A for the LLMs' solutions.

(Cobbe et al., 2021), and the MATH dataset (Hendrycks et al., 2021b), which may no longer provide a true test of an LLM's mathematical capabilities. On the other hand, high-quality, expert-crafted original problems are scarce. For instance, a study by OpenAI (Davis & Aaronson, 2023) included only 105 such problems in high school and university-level science and math.

To directly address these challenges, we introduce the "MathOdyssey" dataset, a rigorously curated collection of 387 mathematical problems for evaluating the general mathematical capacities of LLMs. See examples in Table 1. The MathOdyssey dataset features a spectrum of questions from Olympiad-level competitions, advanced high school curricula, and university-level mathematics. Mathematics professionals, including high-school educators, researchers, and university professors. The key distinction of our dataset is its expert-driven creation, which minimizes the risk of data contamination.

Furthermore, we open-source the MathOdyssey dataset to facilitate its use in evaluating other LLMs. The dataset has not been used for training by LLMs. We explore its utility in benchmarking the advanced mathematical reasoning abilities of LLMs. By ensuring the originality and confidentiality

of the questions, we maintain the integrity and fairness of the assessments, providing a reliable tool for advancing research into artificial general intelligence.

Our contributions are as follows:

- We introduce a new mathematical challenge that provides different levels of mathematical problems and covers a wider range of subject areas.

- We open source the MathOdyssey benchmark dataset, a meticulously curated collection of mathematical problems spanning various domains and levels, complete with natural language solutions. This dataset is specifically designed to probe the reasoning abilities of LLMs, offering a unique tool for assessing AI performance in complex mathematical reasoning. Each question has an objective answer serving as 'ground-truth', allowing for objective evaluation on the LLM outputs. In particular, the Open-Answer problems emphasize the importance of detailed reasoning and solution.

- We conduct a comprehensive benchmark analysis using our dataset on both open-source and closed-source LLMs. Our findings reveal that while closed-source models currently lead, open-source models are rapidly catching up, highlighting the competitive landscape of LLM capabilities in mathematical problem-solving.

## 2 RELATED WORK

**Large Language Models for Mathematics.** Applying large language models (LLMs) to mathematical problems has led to significant strides, though solving such problems remains challenging due to the need for highly complex and symbolic multi-step reasoning capabilities. Both GPT-3.5 and GPT-4 (OpenAI, 2023) have shown promising reasoning abilities for complex mathematical tasks, such as those in the MATH dataset (Hendrycks et al., 2021a). However, the performance of open-source models, like Llama-1 and Llama-2 (Touvron et al., 2023), is still far from satisfactory in this domain. To enhance the mathematical problem-solving abilities of LLMs, prompt-based methods have also been developed (Wei et al., 2022; Wang et al., 2022; Zhou et al., 2022). These methods aim to improve reasoning and accuracy by guiding the models through structured prompts that help in breaking down complex problems into manageable steps.

**Mathematical Evaluation for Large Language Models.** Evaluating the mathematical capacity of large language models (LLMs) is crucial. Benchmarks such as GSM8K (Cobbe et al., 2021), which targets middle-school level mathematics, and MATH (Hendrycks et al., 2021a), which focuses on high-school math competitions, have been widely used. For university-level problems, datasets like ProofNet (Azerbayev et al., 2023a) and OCWCourses (Lewkowycz et al., 2022) are prominent. Additionally, MiniF2F (Zheng et al., 2022) and AlphaGeometry (Trinh et al., 2024) provide Olympiad-level problems, while the SAT dataset (Azerbayev et al., 2023b) includes problems from the College Board SAT examination. These datasets have limitations, particularly at the undergraduate level and above, where they fall short in addressing graduate-level and competition-level difficulties (Frieder et al., 2024). To address this gap, we introduce the MathOdyssey dataset, a diverse collection of mathematical problems designed to serve as a rigorous benchmark for assessing both open-source and closed-source models. Table 2 highlights the properties of MathOdyssey compared to relevant benchmarks, emphasizing the different levels and the diversity of subject areas and question types in our benchmark. This dataset spans a spectrum of difficulty levels, from high school to advanced university mathematics, highlighting the evolving capabilities and ongoing challenges in LLM mathematical problem-solving.

## 3 MATHODYSSEY

To evaluate the mathematical reasoning abilities of LLMs, we create the MathOdyssey dataset, a rigorously curated collection designed by professionals from both universities and high schools. To ensure comprehensive evaluation and promote transparency, we have made the entire MathOdyssey dataset and benchmarking code publicly available. This allows other researchers to replicate our study, compare methods, and explore new approaches using the dataset.

| Dataset | Year | Description | # of Test |
|---------|------|-------------|-----------|
| GSM8k (Cobbe et al., 2021) | 2021 | 8.5k middle-school level math word problems | 1k |
| MATH (Hendrycks et al., 2021b) | 2021 | 12.5k high-school math competitions | 5k |
| OCWCourses (Lewkowycz et al., 2022) | 2022 | University-level, MIT's OpenCourseWare | 272 |
| MiniF2F (Zheng et al., 2022) | 2023 | Olympiad-level | 488 |
| SAT (Azerbayev et al., 2023b) | 2023 | Figureless questions from SAT | 32 |
| ProofNet (Azerbayev et al., 2023a) | 2023 | University-level, proofs | 371 |
| AlphaGeometry (Trinh et al., 2024) | 2024 | Olympiad Geometry only | 30 |
| MathOdyssey (this work) | 2024 | High School, University-level, Olympiad-level | 387 |

Table 2: Comparison of existing evaluation datasets for testing AI in mathematics. These datasets are limited, especially in the availability of high-quality, expert-crafted original problems with varying difficulty levels.

## 3.1 DATA COLLECTION

**Design Principle.** The motivation behind the design of the MathOdyssey dataset is to establish a new benchmark representing the pinnacle of human intellectual achievement, encouraging researchers to push the boundaries of LLMs' mathematical reasoning capabilities. To realize this vision, we have curated challenges that epitomize comprehensive levels of math problems. Specifically, our benchmark includes:

- Inclusion of diverse levels of math problems: Ensuring a comprehensive understanding and catering to various proficiency levels promotes a well-rounded mastery of mathematical concepts and problem-solving skills. This dataset offers a range of problems, starting from basic concepts and gradually increasing in difficulty to cover advanced topics. This allows for a thorough evaluation of AI capabilities across various levels of high school and university mathematics.

- Inclusion of different subject area problems: Enhancing LLMs' mathematical proficiency by exposing them to a wide range of concepts and techniques, from foundational arithmetic to advanced topics such as algebra, number theory, geometry, combinatorics, and calculus. These diverse subject areas help identify LLMs' strengths and areas for improvement, encouraging the development of critical mathematical reasoning, problem-solving skills, and a deeper appreciation for the interconnected nature of mathematics. By integrating various mathematical disciplines, researchers can create a more engaging and comprehensive learning environment that prepares LLMs for complex real-world challenges in mathematics.

- Provision of objective answers and detailed solutions: The objective answers serve as 'ground-truth', allowing for objective evaluation of the LLM outputs. In particular, the Open-Answer problems emphasize the importance of detailed reasoning and solution. Given the varying difficulty and subject areas of these problems, which may exceed comprehension without a specialized background in mathematics, each problem is accompanied by expertly crafted solutions detailing the reasoning steps involved. These solutions are useful for evaluation and can enhance the assessment of LLMs' reasoning processes.

**Human professionals.** The dataset was created by human professionals to ensure high quality. Experts developed a wide range of mathematical problems for the MathOdyssey dataset, featuring a spectrum of questions from Olympiad-level competitions, advanced high school curricula, and university-level mathematics. Mathematics professionals, including high-school educators, university professors, and researchers, crafted these problems. Their involvement ensures the dataset not only supports advanced AGI research but also fosters necessary interdisciplinary collaboration.

A typical problem in the MathOdyssey dataset comprises three components: the problem, the answer, and the reasoning, as detailed in Table 1. The problems are original and not sourced from previous datasets or textbooks. Each problem is accompanied by an answer and a detailed solution that explains the reasoning process used to derive the answer. After creation, the problems undergo independent review by a separate team of researchers with expertise in mathematics. This team assesses the problems and their solutions, eliminating any ambiguous or redundant responses to

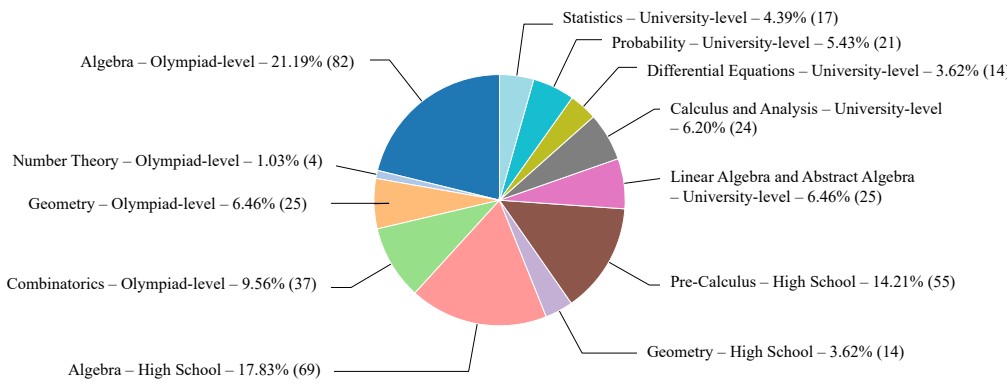

Figure 1: Mathematical problems across educational levels. We curate and categorize problems by difficulty and subject area.

enhance the set's validity and reliability. This rigorous process guarantees the quality and dependability of the final problem set.

### 3.2 DATASET ANALYSIS

To understand the properties of the MathOdyssey dataset, we analyze the questions and answers. Specifically, we explore (i) the difficulty of questions based on the type of reasoning required to answer them, (ii) the subject areas of the problems, and (iii) the diversity of answer types.

**Difficulty of questions.** In the MathOdyssey dataset, each category is designed to evaluate different facets of mathematical reasoning and problem-solving capabilities, ranging from fundamental high school concepts to complex university-level theories, as summarized in Figure 1. This diverse dataset is structured into three distinct levels to challenge various aspects of mathematical knowledge:

- **Olympiad-level**: It tests advanced problem-solving skills with questions in Algebra, Number Theory, Geometry, and Combinatorics.

- **High School**: Broadening the scope, this category includes problems in Algebra, Geometry, and Pre-Calculus, covering a comprehensive range of high school math concepts.

- **University-level**: Catering to higher education, this segment offers challenges in Linear and Abstract Algebra, Calculus and Analysis, Differential Equations, Probability, and Statistics, suitable for university students.

The MathOdyssey dataset categorizes mathematical problems across different educational levels, helping to understand the distribution and scope of problems included in the dataset. For Olympiad-level Competition, the categories and their respective percentages are Algebra (21.19%), Number Theory (1.03%), Geometry (6.46%), and Combinatorics (9.56%), totaling 38.24%. For High School Mathematics, the categories are Algebra (17.83%), Geometry (3.62%), and Pre-Calculus (14.21%), totaling 35.66%. For University-level, the categories are Linear and Abstract Algebra (6.46%), Calculus and Analysis (6.20%), Differential Equations (3.62%), Probability (5.43%), and Statistics (4.39%), totaling 26.10%. Three subject areas, Differential Equations, Probability, and Statistics, only appear at the University level.

**Subject areas of the problems.** The problems encompass a wide range of topics, including Algebra, Number Theory, Geometry, Combinatorics, Pre-Calculus, Linear and Abstract Algebra, Calculus and Analysis, Differential Equations, Probability, and Statistics, as shown in Figure 1. The MathOdyssey dataset encompasses a wide range of subject areas, providing a comprehensive testing ground for the mathematical reasoning and problem-solving capabilities of large language models

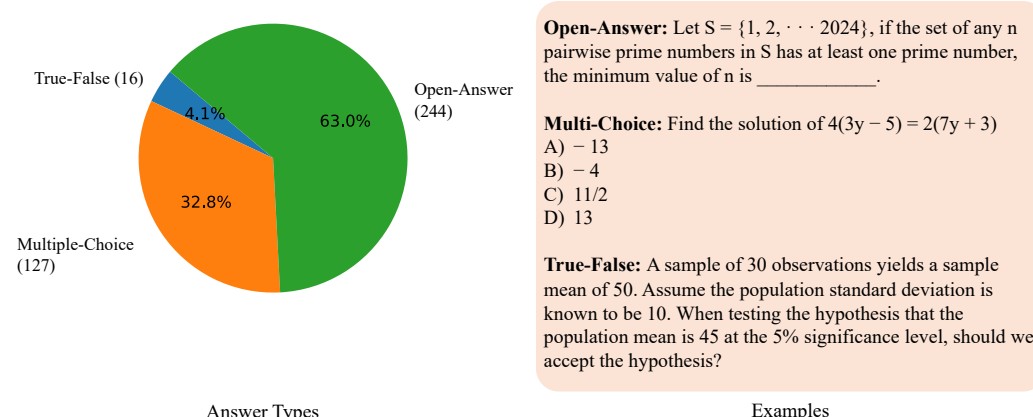

Figure 2: There are three answer-types: True-False questions, Multiple-Choice questions and Open-Answer questions.

(LLMs). Algebra problems constitute 21.19% from Olympiad-level Competition and 17.83% from High School Mathematics, making them the most represented areas in the dataset. In contrast, Number Theory problems, with only 1.03% from Olympiad-level Competition, have the lowest representation. Pre-Calculus problems, accounting for 14.21% of High School Mathematics, play a significant role in preparing students for more advanced calculus topics. Other subject areas, including Calculus and Analysis, Linear and Abstract Algebra, Differential Equations, Probability, and Statistics, each contribute around 4% to 8% to the dataset. See Appendix B for examples that help better understand the reasoning required to answer the questions.

**Diversity of answer types.** The MathOdyssey dataset includes a variety of answer types, providing a comprehensive assessment of the mathematical reasoning and problem-solving capabilities of large language models (LLMs). The distribution of answer types is shown in Figure 2, and it is categorized into three main types: True-False questions, Multiple-Choice questions, and Open-Answer questions. The distribution of answer types in the MathOdyssey dataset is designed to provide a well-rounded evaluation of LLMs' mathematical capabilities. With 62.8% of the questions being open-answer, the dataset emphasizes the importance of detailed reasoning and solution generation. Multiple-choice questions, making up 33.1%, help assess the models' ability to choose correct answers from given options, while true-false questions, at 4.1%, provide a quick check of fundamental understanding. This diverse mix of answer types ensures that LLMs are tested on various aspects of mathematical problem-solving, from basic validation to complex reasoning and solution generation, requiring an understanding of the concepts.

## 4 EXPERIMENTS

Our goal is to provide a comprehensive standardized dataset to evaluate LLMs on mathematical reasoning. By comparing different models, our benchmarks highlight their strengths and weaknesses.

### 4.1 MODELS

We evaluate both open-source and closed-source LLMs. The models tested include GPT-4 o1-preview, GPT-4 Turbo, GPT-4, GPT-3.5 Turbo, Gemini models, Claude 3, and Llama-3-70B. All models are tested using chain-of-thought reasoning (Wei et al., 2022). See Appendix C for details of the baselines and prompts.

## 4.2 MODEL EVALUATION

A key advantage of the MathOdyssey data is that every question has an objective answer, so that it is straightforward to check the correctness by code. Such objective answers avoid subjective judgments from humans, making the evaluation consistent and reliable.

We use GPT-4 to assist in evaluating model accuracy, particularly for open-answer questions. The metric measures the similarity between the predicted and ground truth answers. In the MathOdyssey dataset, various types of questions and answers are included. We employ a prompt-based method to provide scores for evaluation, considering the following criteria:

- Mathematical Equivalence: Verify answers based on mathematical equivalence using advanced tools like symbolic computation software to confirm the equivalence of different algebraic or symbolic expressions.

- Scoring: Assign a score of '1' for answers that match or are equivalent to the provided solution (exact value, choice label, or correctly rounded numerical approximation). Assign a score of '0' for incorrect answers without providing explanatory feedback.

- Handling Multiple Choices: Consider the answer correct if the student correctly identifies the choice that matches the solution. Also, treat the corresponding choice as correct if the student provides the exact value that aligns with the problem's context.

- Numerical Equivalence: Accept numerical answers that are correct to at least two decimal places or more, depending on the required precision.

- Symbolic and Algebraic Identities: Recognize and accept equivalent algebraic forms as correct, such as standard mathematical identities.

- Trigonometric and Logarithmic Forms: Accept equivalent trigonometric and logarithmic expressions, acknowledging transformations that change the form but not the value.

- Comprehensive Evaluation: Encourage the use of computational tools for checking equivalence in cases where expressions are too complex for straightforward visual inspection.

See Appendix D for the requirements and prompts used in the evaluation method. We have also made our evaluation code accessible to the public via our GitHub repository, including not only the code but also detailed documentation and usage examples.

## 4.3 RESULTS AND ANALYSIS

We first report the performance on our mathematical benchmarks, as shown in Table 3. Our observations indicate that the benchmark is challenging for these models, with overall performance below 60% except for GPT-4 o1-preview.[2] The recent GPT-4 o1-preview achieves the highest overall performance at 65.12%, demonstrating that incorporating chain-of-thought learning significantly enhances capabilities. The Gemini Math-Specialized 1.5 Pro also performs well, ranking second with a score of 55.8%, suggesting that specialized training can further improve specific skill areas. GPT-4 Turbo achieves 49.35%, followed by Gemini 1.5 Pro at 45.0%, and Claude 3 Opus at 40.6%, all showing competitive performance. For closed-source models (specifically the GPT series) and state-of-the-art open-source models such as Llama-3, the results show that the selected open-source models not only surpass the performance of GPT-3.5 but are also approaching the capabilities of earlier versions of GPT-4.

When comparing different levels of mathematical problems for GPT models, we observe that High School mathematics is the easiest category for all models, with GPT-4 models scoring above 70%. Olympiad-level problems are the most difficult, with all models scoring below 11% except for GPT-4 o1-preview. Similar trends are seen for Llama-3-70B, with their performance in the Olympiad-level category being even lower, at less than 9%.

Furthermore, closed-source models, particularly the GPT-4 o1-preview and GPT-4 Turbo, exhibit stronger performance in high school and university-level math, highlighting ongoing advancements

---

[2]Advanced prompting methods using GPT-4 models in the contest have achieved performance improvements between 60% and 70%.

| Model | Olympiad-level | High School | University-Level | Overall |
|---|---|---|---|---|
| GPT-4 o1-preview | 45.27% | 79.71% | 74.26% | 65.12% |
| GPT-4 Turbo | 10.81% | 84.06% | 58.42% | 49.35% |
| GPT-4 | 5.41% | 85.51% | 44.55% | 44.19% |
| GPT-3.5 Turbo | 3.38% | 39.13% | 16.83% | 19.64% |
| Gemini | | | | |
|   -1.5 Pro | - | - | - | 45.0 % |
|   -Math-Specialized 1.5 Pro | - | - | - | 55.8 % |
| Claude 3 Opus | - | - | - | 40.6 % |
| Llama-3-70B | 8.78% | 73.19% | 24.75% | 35.92% |

Table 3: Results for different LLMs. The performance of Gemini 1.5 Pro and Claude 3 Opus are quoted from the Gemini 1.5 report (Reid et al., 2024). Both GPT-4-Turbo and Gemini 1.5 Pro outperform the other models. For GPT-4-Turbo, we use results based on gpt-4-turbo-2024-04-09. For GPT-4, we use results based on gpt-4-0125. For GPT-3.5 Turbo, we use results based on gpt-3.5-turbo-0125.

| Category | GPT-4 o1-preview | GPT-4 Turbo | GPT-3.5 Turbo | Llama3-70b |
|---|---|---|---|---|
| **Olympiad-level:** | | | | |
| Algebra | 51.22% | 12.20% | 3.66% | 9.76% |
| Number Theory | 75.00% | 0.00% | 0.00% | 0.00% |
| Geometry | 56.00% | 4.00% | 8.00% | 4.00% |
| Combinatorics | 21.62% | 13.51% | 0.00% | 10.81% |
| **High School Mathematics:** | | | | |
| Algebra | 81.16% | 85.51% | 39.13% | 78.26% |
| Geometry | 92.86% | 85.71% | 50.00% | 85.71% |
| Pre-Calculus | 74.47% | 80.85% | 34.04% | 68.09% |
| **University-level:** | | | | |
| Differential Equations | 71.43% | 64.29% | 35.71% | 64.29% |
| Linear & Abstract Algebra | 92.00% | 72.00% | 12.00% | 20.00% |
| Calculus & Analysis | 79.17% | 70.83% | 16.67% | 33.33% |
| Probability | 52.38% | 23.81% | 0.00% | 0.00% |
| Statistics | 70.59% | 58.82% | 29.41% | 17.65% |

Table 4: Results for different LLMs across various subject areas. Note that the results are used for evaluating the LLMs by direct comparison and may be improved with different prompting methods.

in their development. This data underscores the rapid progression of closed-source models in handling increasingly difficult mathematical questions over time. The performance gap between the previous closed-source model, GPT-4 Turbo, and the open-source Llama-3 for difficult mathematical problems is notably narrow. However, the gap between recent closed-source model GPT-4 o1-preview becomes larger. For instance, except that GPT-4 o1-preview achieves 45.27%, GPT-4 Turbo achieves an overall accuracy of 10.81% in the Olympiad-level mathematics, while Llama-3 achieves 8.78%. This demonstrates that both models, despite notable progress, still face significant challenges in solving these complex problems. However, for other difficulty levels, the gap becomes larger. For example, GPT-4 Turbo achieves 84.06% in high school mathematics, while Llama-3-70B scores only 73.19%, a difference of more than 10%.

Table 4 presents the performance of various LLMs across different subject areas. GPT-4 o1-preview consistently outperforms others, particularly excelling in Olympiad-level subjects such as Algebra, Number Theory, Geometry, and Combinatorics, as well as university-level subjects like Differential Equations, Linear & Abstract Algebra, Calculus & Analysis, and Statistics. GPT-4 Turbo follows with the second-best performance. GPT-3.5 Turbo demonstrates steady but lower performance compared to GPT-4 Turbo. Llama-3-70B performs better than GPT-3.5 Turbo in some areas, notably High School Mathematics, including Algebra and Geometry. However, it struggles in university-level subjects like Linear & Abstract Algebra, Calculus & Analysis, and Probability when compared to GPT-4 o1-preview and GPT-4 Turbo.

## 5 CONCLUSION

We introduce MathOdyssey, a dataset for assessing LLMs' mathematical problem-solving skills. Our dataset, evaluation methods, and code are openly available. We have shown that while LLMs, both open-source like Llama-3, and closed-source such as the GPT series, demonstrate proficiency in routine and moderately difficult mathematics, they struggle significantly with complex Olympiad-level problems. Additionally, we have revealed promising developments; open-source models are beginning to approach the performance levels of earlier GPT-3.5 iterations. Despite this progress, performance on the most challenging questions remains low, highlighting a clear gap that future advancements need to address.

Ultimately, our research underscores the ongoing journey towards achieving human-like mathematical reasoning in AI, with the MathOdyssey dataset serving as a benchmark for catalysing future developments. We are optimistic that continued research will progressively bridge the existing capability gap. In the future, expanding the MathOdyssey dataset to include a wider range of problem types and enhancing metrics to better capture deep mathematical reasoning can yield further insights into LLM capabilities.

**Limitation.** While the MathOdyssey dataset includes a variety of problems across different levels of mathematics, the questions may not cover all types of mathematical reasoning or problem-solving approaches. This limitation could affect how well the dataset generalizes to other forms of mathematical challenges not represented in your collection.

**Future.** To address generalizability limitations, future work involves expanding the dataset to include a wider range of mathematical topics and problem types, including those that require visual representations, proofs, or interactive problem-solving.

## DATA COPYRIGHT AND ETHICS STATEMENT

The MathOdyssey dataset comprises a range of problems from Olympiad-level competitions, advanced high school curricula, and university-level mathematics, created by mathematics professionals, including high school educators, researchers, and university professors. We retain the copyright for these problems and are pleased to distribute the dataset under the "CC BY-SA 4.0" license.

## REPRODUCIBILITY STATEMENT

The dataset, results, and evaluation code are publicly available at https://anonymous.4open.science/r/mathodyssey-C587/.

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

## A  MATHODYSSEY DATASET EXAMPLES AND SOLUTIONS

**Olympiad-level Problem.** Let $S = \{1, 2, \cdots 2024\}$, if the set of any $n$ pairwise prime numbers in $S$ has at least one prime number, the minimum value of $n$ is ___________.

**Answer:** 16

**Reasoning:** Taking the 15 numbers $1, 2^2, 3^2, ..., 43^2$ violates the condition. Furthermore, since $S$ does not contain any non-prime numbers with a minimum prime factor of at least 47, there are only 14 types of non-prime numbers in $S$, excluding 1. Applying the Pigeonhole Principle, we conclude that $n = 16$.

**GPT-4 Turbo's Solution:** Answer: 307

Reasoning: The minimum number of elements required such that any subset of these elements from the set $S = \{1, 2, ..., 2024\}$ containing pairwise prime numbers includes at least one prime number is 307. This is based on the count of prime numbers up to 2024 (approximately 306) and the application of the Pigeonhole Principle, which dictates that selecting more numbers than there are primes forces the inclusion of a prime number or a violation of the pairwise prime condition.

**High School Problem.** What are the solutions of the quadratic equation $15x^2 = 2x + 8$?

$$\text{A)} \{-\frac{4}{3}, -\frac{3}{2}\} \quad \text{B)} \{-\frac{4}{5}, \frac{2}{3}\} \quad \text{C)} \{-\frac{3}{2}, \frac{4}{5}\} \quad \text{D)} \{-\frac{2}{3}, \frac{4}{5}\}$$

**Answer:** $D$

**Reasoning:** First move all terms to one side: $15x^2 - 2x - 8 = 0$. Then factor into $(5x-4)(3x+2) = 0$. Setting $5x - 4$ to zero results in a solution of $x = \frac{4}{5}$ and setting $3x + 2$ to zero results in a solution of $x = -\frac{2}{3}$.

**GPT-4 Turbo's Solution:** Answer: $D$

Reasoning: The quadratic equation was first rearranged to standard form, coefficients were identified, and the quadratic formula was applied to find the solutions. The solutions were verified by substituting back into the original equation.

**University-level Problem.** Find the limit

$$\lim_{x \to 1} \frac{f(2x^2 + x - 3) - f(0)}{x - 1}$$

given $f'(1) = 2$ and $f'(0) = -1$.

**Answer:** $-5$

**Reasoning:** Let $g(x) = 2x^2 + x - 3$. Since $g(1) = 0$, the desired limit equals $\lim_{x \to 1} \frac{f(g(x)) - f(g(1))}{x-1}$. By the definition of the derivative and the chain rule and noting that $g'(1) = 5$, we have

$$\lim_{x \to 1} \frac{f(g(x)) - f(g(1))}{x - 1} = f'(g(1))g'(1) = f'(0)g'(1) = (-1)(5) = -5.$$

**GPT-4 Turbo's Solution:** Answer: $-5$

Reasoning: The limit was calculated by interpreting it as the derivative of a composed function, applying the chain rule, and substituting the given derivative values.

## B  MATHODYSSEY DIFFERENT SUBJECT AREAS

Table 5 presents MathOdyssey examples spanning various subject areas. These encompass Algebra, Number Theory, Geometry, Combinatorics, Pre-Calculus, Linear and Abstract Algebra, Calculus and Analysis, Differential Equations, as well as Probability and Statistics.

## C  BASELINES AND PROMPTS

Figure 3 depicts the prompt utilized for guiding Language Models (LLMs) in solving mathematical problems within our experimental framework. This prompt distinctly outlines the system's role as a math professor, delineating task specifications and the anticipated output format for tackling intricate mathematical challenges.

| Subject Area | Example |
|---|---|
| Algebra | Let $S = \{1, 2, \cdots 2024\}$, if the set of any $n$ pairwise prime numbers in $S$ has at least one prime number, the minimum value of $n$ is __________. |
| Number Theory | A natural number whose last four digits are 2022 and is divisible by 2003 has a minimum value of __________. |
| Geometry | In a cube $ABCD - A_1B_1C_1D_1$, $AA_1 = 1$, $E$, $F$ are the midpoints of edges $CC_1, DD_1$, then the area of the cross-section obtained by the plane AEF intersecting the circumscribed sphere of the cube is __________. |
| Combinatorics | If three points are randomly chosen from the vertices of a regular 17-sided polygon, what is the probability that the chosen points form an acute-angled triangle? |
| Pre-Calculus | In $\triangle ABC$, $AB = 10$ cm, $\angle B = 90°$, and $\angle C = 60°$. Determine the length of BC. 

 A) 10 cm  B) $10\sqrt{3}$ cm  C) $\dfrac{10\sqrt{3}}{3}$ cm  D) 20 cm |
| Linear and Abstract Algebra | Find the solution $[x_1, x_2, x_3]$ to the following equations 

 $\begin{cases} x_1 + 3x_2 + 3x_3 = 16, \\ 3x_1 + x_2 + 3x_3 = 14, \\ 3x_1 + 3x_2 + x_3 = 12. \end{cases}$ |
| Calculus and Analysis | Evaluate the following limit: 

 $\lim\limits_{n \to \infty} \left( \sqrt{n^2 + 2n - 1} - \sqrt{n^2 + 3} \right).$ |
| Differential Equations | Consider the differential equation $\frac{dy}{dx} = xy$. Find the value of $y(\sqrt{2})$ given that $y(0) = 2$. |
| Probability | Suppose that $A$, $B$, and $C$ are mutually independent events and that $P(A) = 0.2$, $P(B) = 0.5$, and $P(C) = 0.8$. Find the probability that exactly two of the three events occur. |
| Statistics | Given the data set $\{3, 7, 7, 2, 5\}$, calculate the sample mean $\mu$ and the sample standard deviation $\sigma$. Present the answer as $[\mu, \sigma]$. |

Table 5: Examples of different subject areas.

# D  EVALUATION

Figure 4 depicts the prompt employed during the evaluation of large language models in our experiments. This prompt defines the system's role as a math teacher, providing both assessment criteria and the expected output format for grading mathematical problems.

You are now assuming the role of a math professor. Your task is to assist the user by solving complex mathematical problems in a detailed and step-by-step manner.

## Task Requirements:
1. **Detailed Problem Analysis**: Start by analyzing the given problem. Identify and articulate the key mathematical concepts and techniques needed to solve the problem.
2. **Step-by-Step Solution**: Decompose the problem into manageable steps. Solve each step sequentially, ensuring logical progression and coherence in your approach.
3. **Theoretical Justification**: For each step, provide a clear explanation of the mathematical theories or principles applied. Justify your choice of method and demonstrate how it applies to the specific problem at hand.
4. **Calculation Verification**: After solving each step, verify your calculations. Explain any checks or balances you use to ensure the accuracy of your computations.
5. **Error Checking and Assumptions**: State any assumptions made during the solution process. Discuss potential errors or alternative methods that could impact the solution.
6. **Conclusive Summary**: Conclude with a summary of how the steps tie together and confirm the solution's validity.

## Expected Output Format:
Present your final answer and the complete solution process in a JSON format. This should include:
    - A `float` value or a mathematical algebraic expression for the answer.
    - Detailed reasoning for each step of the solution.

Your output should be formatted as a JSON object enclosed in Markdown code blocks tagged with 'json'. For example:

```json
{{
    "reasoning": "<detailed solution process>",
    "answer": "<answer>"
}}
```

Ensure that all task requirements are meticulously followed in your response.

Figure 3: Mathematical problem-solving prompts employed by LLMs.

Assume the role of a math teacher tasked with evaluating student responses against the provided solutions, which may include exact values, multiple-choice answers, or numerical approximations. The question is provided as: {question}, the correct answer is provided as: {true}.

## Evaluation Criteria:
1. **Mathematical Equivalence**: Evaluate answers based on deep mathematical equivalence, not just numerical accuracy. Use advanced tools or techniques to verify if different algebraic or symbolic expressions are equivalent. Tools like symbolic computation software (e.g., Wolfram Alpha, SymPy) should be used to confirm equivalences such as \\( \\frac{{\\sqrt{{6}}-\\sqrt{{2}}}}{{2}} \\) being equivalent to \\( \\sqrt{{2 - \\sqrt{{3}}}} \\).
2. **Scoring**: Assign a score of '1' for any answer that matches or is equivalent to the provided solution, whether it is an exact value, a choice label (e.g., A, B, C), or a correctly rounded numerical approximation. Assign a score of '0' for incorrect answers. Do not provide any explanatory feedback in your evaluation.
3. **Handling Multiple Choices**: If the solution provided is a choice (e.g., A, B, C, D, E, F) and the student identifies this choice correctly, treat it as correct. If the solution is an exact value and the student provides the corresponding choice that reflects this value correctly according to the problem's context, also treat it as correct.
4. **Numerical Equivalence**: Treat numerical answers as equivalent if they are correct to at least two decimal places or more, depending on the precision provided in the solution. For instance, both 0.913 and 0.91 should be accepted if the solution is accurate within two decimal places.
5. **Symbolic and Algebraic Identities**: Recognize and accept equivalent algebraic forms, such as \\( \\sin^2(x) + \\cos^2(x) = 1 \\) or \\( e^{{i\\pi}} + 1 = 0 \\), as correct.
6. **Trigonometric and Logarithmic Forms**: Accept equivalent trigonometric and logarithmic expressions, acknowledging identities and transformations that might alter the form but not the value.
7. **Comprehensive Evaluation**: Encourage the use of computational tools to check for equivalence in cases where expressions are too complex for straightforward visual inspection.

## Expected Output Format:
Present your final answer with a score of '1' or '0' only. Do not include any additional information or feedback in your response.

Please evaluate the student's response with precision, utilizing computational resources as necessary to ensure accurate and fair grading.

Figure 4: Evaluation prompts.

