# OpenReview forum: "MathOdyssey: Benchmarking Mathematical Problem-Solving Skills in Large Language Models Using Odyssey Math Data"
_ICLR.cc/2025/Conference — ICLR 2025 Conference Withdrawn Submission_

### Official Review · Reviewer_KoNW · 2024-10-31

**Soundness:** 1
**Presentation:** 2
**Contribution:** 2
**Rating:** 3
**Confidence:** 4

**Summary:**

This paper presents a new dataset called MathOdyssey, which aims to evaluate the reasoning abilities of large language models (LLMs). The dataset consists of 387 problems, including 148 at the Olympic level, 101 at the university level, and 138 at the high school level. The problems cover several subjects with three answer formats: true/false, multiple choice, and open answers.

The authors evaluate LLMs' math reasoning performance on MathOdyssey using GPT-4 as the answer judger in a zero-shot manner, providing it with specific instructions. They conduct their experiments with seven closed-source LLMs and one open-source model, Llama-3-70B. Their findings reveal that the Llama-3-70B model still falls short when tackling more complex problems.

**Strengths:**

The proposed MathOdyssey dataset is novel and may be somewhat useful to certain researchers. While it introduces a variety of challenging problems, the experimental results provide a rough indication of the performance of closed-source LLMs. Overall, the dataset and findings suggest potential (but limited) usefulness to offer insights into LLM capabilities in mathematical reasoning, albeit with room for more comprehensive analysis.

**Weaknesses:**

MathOdyssey **offers no clear advantages over existing benchmarks**, which may limit the usefulness and contribution of this paper.

- Compared to existing datasets, MathOdyssey is **limited in size**, containing only 387 problems, whereas datasets like GSM8K and MATH include 1,319 and 5,000 problems, respectively. This limitation might impact the reliability of accuracy in ranking the mathematical reasoning abilities of different LLMs.

- **The "difficulty levels" within MathOdyssey are not well-defined**. Although it claims to cover comprehensive levels of math problems, it includes only three educational stages. In contrast, the MathBench [1] dataset offers a wide range of problems, spanning from primary school to university level. By the way, some datasets define the difficulty level as a rating (e.g. an integer number)

- While the authors claim to have diversified answer types, MathOdyssey **only encompasses three distinct answer types**. OlympiadBench [2], however, incorporates a more fine-grained variety of answer types.

- Although MathOdyssey includes several subjects, **the number of testing examples within each subject is relatively small, with many subjects containing fewer than 10 examples**. This limitation may lead to inaccurate analyses across different subjects.

**The experiments are not comprehensive and compelling**:

- The evaluation process is flawed because the authors use GPT-4 as the judge for answers in a zero-shot manner. However, **it is unclear how often this judgment aligns with human evaluators**. An analysis of judgment errors is necessary, and I recommend considering rule-based matching.

- They **include only one open-source LLM, Llama-3-70B** in experiments, which is not comprehensive. The authors should include more open-source LLMs, including both general-purpose chat models and math-specialized LLMs.

- The results and analysis are not compelling, as the reliability of GPT-4's judgment is uncertain. Additionally, **conducting an error analysis could provide valuable insights into how LLMs get wrong in solving math problems**.


This paper **is poorly written** and either **lacks important details or makes inaccurate claims or claims without proper citations** (see Questions).


[1] Liu, H., Zheng, Z., Qiao, Y., Duan, H., Fei, Z., Zhou, F., ... & Chen, K. (2024). MathBench: Evaluating the Theory and Application Proficiency of LLMs with a Hierarchical Mathematics Benchmark. arXiv preprint arXiv:2405.12209.

[2] He, C., Luo, R., Bai, Y., Hu, S., Thai, Z. L., Shen, J., ... & Sun, M. (2024). Olympiadbench: A challenging benchmark for promoting agi with olympiad-level bilingual multimodal scientific problems. arXiv preprint arXiv:2402.14008.

**Questions:**

Questions and Concerns:

- Line 103-105: The authors claim, "The key distinction of our dataset is its expert-driven creation, which minimizes the risk of data contamination." In lines 106-107, they state, "The dataset has not been used for training by LLMs." To substantiate these claims, the authors should conduct a contamination detection analysis, which has not been done.

- The term "open-answers" is mentioned without a clear definition. Can these open-answers be further classified into more fine-grained types, similar to the classification in OlympiadBench [1]?

- Given that GPT-4 is used as the answer judge for all experiments and analyses, it is necessary to conduct human evaluations to determine the reliability of GPT-4 as a judge.

- The evaluation prompt requires output in JSON format. However, some LLMs are fine-tuned to produce answers delimited by "boxed". Have you investigated whether LLMs conform to the expected format?

- Have you considered incorporating difficulty levels like MATH? Some high school problems might be easier than certain university-level problems.

- Results on More Open-Source LLMs: Can you provide results from more open-source LLMs?

Other Issues:

Lack of Appropriate Citations:

- Lines 39-41: "has achieved more than a 90% success rate"

- Lines 44-45: "technological advancement but a crucial step toward developing more general and capable artificial intelligence systems"

- Lines 51-52: "Moreover, a significant obstacle is that many existing datasets might have been included in the training phases of these models, potentially skewing performance metrics."

- Lines 320-322: Citations or links for LLMs are missing.

Inaccurate Statements:

- Lines 47-49: "it remains uncertain how well they handle more complex mathematical challenges, such as those found in university-level courses and competitive high school mathematics." In fact, datasets like MATH and MathBench [2] include competitive high school mathematics and university-level courses, respectively.

- Lines 51-52: "Moreover, a significant obstacle is that many existing datasets might have been included in the training phases of these models, potentially skewing performance metrics." There are, however, many variants of existing datasets that address this issue, such as GSM-1K [3], which also weakens the contribution of this paper because MathOdyssey is limited in size.

- Table 2: The number of examples for GSM8K should be 1319.

[1] He, C., Luo, R., Bai, Y., Hu, S., Thai, Z. L., Shen, J., ... & Sun, M. (2024). Olympiadbench: A challenging benchmark for promoting agi with olympiad-level bilingual multimodal scientific problems. arXiv preprint arXiv:2402.14008.

[2] Liu, H., Zheng, Z., Qiao, Y., Duan, H., Fei, Z., Zhou, F., ... & Chen, K. (2024). MathBench: Evaluating the Theory and Application Proficiency of LLMs with a Hierarchical Mathematics Benchmark. arXiv preprint arXiv:2405.12209.

[3] Zhang, H., Da, J., Lee, D., Robinson, V., Wu, C., Song, W., ... & Yue, S. (2024). A careful examination of large language model performance on grade school arithmetic. arXiv preprint arXiv:2405.00332.

---

### Official Review · Reviewer_pPVq · 2024-11-01

**Soundness:** 3
**Presentation:** 3
**Contribution:** 3
**Rating:** 6
**Confidence:** 5

**Summary:**

The paper aims to investigate and understand LLMs’ strong problem-solving abilities. The paper introduces a novel dataset “MathOdyssey”. The dataset includes a diverse set of mathematical programs at three levels: High school, university and olympiad level. Each category has a wide range of different problem areas such as Algebra, Number Theory, calculus etc. The dataset contains a total of 387 data points and has novel problems created by mathematics professionals, including high school educators, university professors and researchers. Each problem is accompanied by its final answer and its reasoning chain. They also did a comprehensive evaluation of the dataset and tested it on both closed and open models. They used GPT-4 to assist in evaluating the model accuracy, as the dataset contained a wide range of answer types (open answer, MCQ, and true-false). The evaluation shows that the closed source model particularly GPT-4, o1 and GPT-4 Turbo shows strong performance in high school and university-level math. For open-source models such as Llama-3, the results show that the selected open-source models only surpass the performance of GPT3.5 but are also approaching the capabilities of the earlier version of GPT-4.

**Strengths:**

1. Release of a novel dataset that will help the community as this dataset has not been used in the training of existing models.
2. Comprehensive benching of different models, highlighting their efficiency in solving different categories/areas of problems.
3. Effective use of GPT-4 for evaluating the accuracy of models. The author employs a prompt-based method and provides scores of various categories.

**Weaknesses:**

1. Even though the dataset provides various categories of questions in different areas, the count of individual categories is very small. For example Number Theory – Olympiad-level accounts for only 4 problems, Differential Equations – University-level for 14 problems etc. So do the authors have any plan to extend the count of problems in these areas?
2. Even though the dataset does not use any existing problems, a sanity check for data contamination should be done. Experiments from the paper [1] should be added to ensure no data contamination.
3. Evaluating the dataset using models fine-tuned specifically for solving math problems, such as MathCoder [2] helps show how models trained specifically to solve math problems perform on MathOdyssey.

Reference:

[1] : Golchin, Shahriar, and Mihai Surdeanu. "Time travel in llms: Tracing data contamination in large language models." arXiv preprint arXiv:2308.08493 (2023).

[2] : Wang, Ke, et al. "Mathcoder: Seamless code integration in llms for enhanced mathematical reasoning." arXiv preprint arXiv:2310.03731 (2023).

**Questions:**

Address the weakness of the paper

---

### Official Review · Reviewer_Es1n · 2024-11-03

**Soundness:** 3
**Presentation:** 2
**Contribution:** 4
**Rating:** 6
**Confidence:** 3

**Summary:**

The authors identify a need for a mathematics benchmark that spans a wider breadth of topics and difficulties. They propose MathOdyssey, a benchmark that includes hand-written and curated high-school, university, and Olympic-level problems. Each problem has a unique expected answer and detailed reasoning to aid LLM assessment. They demonstrate that the benchmark is not saturated since GPT-4 o1-preview archives ~65% overall. Further, their wide coverage of topics enables the identification of problem topics for LLMs enabling researchers to focus on those areas.

**Strengths:**

+ The proposed benchmark should have a unique answer enabling easier verification of the correct answer.
+ The problems are crafted specifically for the benchmark avoiding their presence in the pre-training data for LLMs.
+ The problem space covered in terms of topics and difficulty is wide, allowing the identification of problem areas for further research as well as "solved" areas if a topic saturates.

**Weaknesses:**

- Some of the paper language is hyperbolic, for example, S3.1, "Design Principle." Paragraph, L178: "representing the pinnacle of human intellectual achievement" is very strong language and am uncertain the authors could substantiate such a claim unless it is an opinion. Another example is S3.1 L234: "This rigorous process": The curation process, while I can trust was done rigorously, is not presented in sufficient detail for me to make that assessment, and it would be better to instead tone it down to "This process facilitates the quality and dependability...". The paper in general would benefit from a pass that tones down the hyperbolic language to instead focus on the proposed advancements in an objective tone.
- L256-264 could be replaced with "Fig. 1 shows the detailed information". and L266-268 repeat the same information that is already in the figure and could be pointed to.
- The benchmark claims easy verifiability by code but uses GPT-4 as an evaluator (with the associated errors this induces even if the prompt enables the use of tools).
- The ease of having a unique answer is counter-balanced by the false positives induced by correct-answer-with-wrong-reasoning.

**Questions:**

1. Is there a particular reason that an LLM is used as an evaluator instead of using SymPy or Mathematica directly when needed (for example equivalence of symbolic expressions)?
2. While I have not deep dove into the dataset, the example from Table 1 has implicit or missing steps in the university example to my reading. It seems to have an implicit application of L'Hôpital's Rule before taking the limit and then applying the chain rule.
Thus, my question is how did you balance implicit vs explicit steps in the reasoning?
I would expect/hope an LLM to list all steps, including those "obvious" to a human marker, and err on the verbose side; however, I accept this as a personal view/bias and am keen to hear what the dataset prerogative was.

---

### Official Review · Reviewer_c8ZY · 2024-11-04

**Soundness:** 2
**Presentation:** 2
**Contribution:** 2
**Rating:** 3
**Confidence:** 3

**Summary:**

The manuscript presents an original and challenging dataset for mathematical problem-solving, encompassing various subjects and difficulty levels. Then the paper conducts comprehensive examinations on both open-source and closed-source LLMs. The findings reveal that while closed-source models currently lead, open-source models are rapidly catching up, highlighting the competitive landscape of LLM capabilities in mathematical problem-solving.

**Strengths:**

* The paper is well-motivated, as GPT-4o poses a significant challenge to current mathematical benchmarks. The introduction and open-sourcing of high-quality, difficult mathematical problems is a meaningful contribution to the field.

* The dataset features distinct levels of difficulty and sub-domain classifications, which enhance its uniqueness.

**Weaknesses:**

1. The manuscript lacks coverage of important related work and further clarification on the difference and improvements compared to them.
    * OlympiadBench: A Challenging Benchmark for Promoting AGI with Olympiad-Level Bilingual Multimodal Scientific Problems
    * Omni-MATH: A Universal Olympiad Level Mathematic Benchmark for Large Language Models
    * OlympicArena: Benchmarking Multi-discipline Cognitive Reasoning for Superintelligent AI
    * Have LLMs Advanced Enough? A Challenging Problem Solving Benchmark For Large Language Models

2. There are missing baseline comparisons that are crucial for evaluating the open-source models on the proposed dataset, like Qwen2.5-MATH, DeepSeek-Coder, and so on.

3. It is unclear how the authors ensure that the data has not been previously encountered. If the problems are original, details regarding the creation principles and methodologies should be included in the paper. Additionally, how is the correctness of answers verified? Have the authors conducted cross-validation or sampling tests to ensure reliability? What is the accuracy rate?
4. The conclusions drawn seem predictable and do not provide substantial insights. Are there fine-grained analyses and interesting findings?

**Questions:**

1. While the authors mention diverse answer types, there are only three categories presented. Moreover, there seems to be a discrepancy between the data in Figure 2 and that described in line 304. Given that existing mathematical models have undergone extensive pre-training and can effectively comprehend natural language queries, why does the diversity of answer types pose a greater challenge for these models? Is there a variance in accuracy across different question types? Do models exhibit inconsistencies when dealing with various types of questions?
2. The creation details of the problems:
* Please include principles and details of problem creation in an appendix. How is answer correctness ensured? Have cross-validation or sampling methods been employed? What is the accuracy rate?
* How are problems categorized into different difficulty levels?
* Does an individual problem encompass multiple domains simultaneously? How is this handled?
3. For evaluation, the authors directly employed GPT-4 for assessment. How to ensure the accuracy of GPT-4's evaluations, and what is the consistency with human assessments? Furthermore, it is recommended that the authors incorporate rule-based evaluation methods to enhance accessibility.

---

### Note · Authors · 2024-12-06

I have read and agree with the venue's withdrawal policy on behalf of myself and my co-authors.